# The Relationship between Late-Life Depression and Cognitive Function in Older Korean Adults: A Moderation Analysis of Physical Activity Combined with Lower-Body Muscle Strength

**DOI:** 10.3390/ijerph19148769

**Published:** 2022-07-19

**Authors:** Jiyoung Kong, Minjeong Kang, Hyunsik Kang

**Affiliations:** College of Sport Science, Sungkyunkwan University, Suwon 16419, Korea; paradose77@naver.com (J.K.); kangmin125@skku.edu (M.K.)

**Keywords:** physical activity, physical fitness, mental health, older adults

## Abstract

Background: This study examined the relationship of physical activity (PA) combined with lower-body muscle strength to late-life depression and cognitive impairment in 10,097 participants (6062 females) ≥ 65 years of age. Methods: Exposures were PA, sit-to-stand test (STST), and depressive symptoms. Outcome was cognitive performance. Results: Depressed individuals had an increased risk of mild cognitive impairment (MCI; odds ratio (OR), adjusted OR = 1.845 and 95% confidence interval (CI) = 1.580–2.154, *p* < 0.001) compared with non-depressed individuals. Individuals who had insufficient PA and a poor STST and either insufficient PA or a poor STST had an increased risk of MCI (adjusted OR = 1.329 and 95% CI = 1.209–1.46, *p* < 0.001 and adjusted OR = 2.822 and 95% CI = 2.488–3.200, *p* < 0001, respectively) compared with individuals who had sufficient PA and a good STST. A significant moderation effect of PA combined with lower-body muscle strength on the relationship between depression and cognitive function was observed (β = −1.3923; 95% CI = −2.1505 to −0.6341, *p* < 0.003). Conclusions: The negative effect of late-life depression on cognitive function was incremental in the order of sufficient PA and a good STST, insufficient PA or a poor STST, and insufficient PA and a poor STST.

## 1. Introduction

Aging of the global population is an inevitable trend attributable to a decrease in fertility rates and an increase in life expectancy, resulting in numerous geriatric conditions [1]. South Korea officially reached an aging society status in 2000—with 7.2% of the population ≥ 65 years of age—and aged society status in 2017, with older adults comprising 15.5% of the population. The country is now projected to become a super-aged society by 2025 [2]. Consequently, many elderly Korean individuals are currently or will be at risk of developing geriatric syndromes, including chronic diseases, functional and physical limitations, frailty, cognitive impairment, and dementia [3,4].

Age-related cognitive decline is a mental condition that should receive special attention, as it may reflect early clinical symptoms of dementia [5,6]. The prevalence of dementia among elderly Koreans is estimated to be 9.2%, which is higher than Western and other Asian populations [7], and is likely to increase due to a rapid rise in the older population [8]. Therefore, understanding the etiology of age-related cognitive decline is imperative for the establishment of a health policy to slow cognitive impairment and minimize the odds of its progress toward dementia.

Late-life depression—which is defined as depression that occurs for the first time after age 60 [9]—is another mental illness with significant consequences, including suicidal ideation [10] and dementia [11]. Depression is intertwined with cognitive impairment [12] and dementia [13] as a prodromal symptom. Furthermore, cognitive impairment and depression are likely to co-occur in elderly people [14] and to escalate after 70 years of age, emphasizing the clinical importance of assessing cognitive functioning in older adults with depressive symptoms [15].

Physical activity (PA) is positively correlated with brain processing speed [16], hippocampal neurogenesis [17], and neural plasticity [18], providing a protective effect against mental illnesses such as late-life depression, age-related cognitive decline, and dementia [17]. The cognitive benefits of PA were reported in studies involving older Chinese [19] and Swedish adults [20]. The anti-depressant effects of PA are also summarized in clinically depressed patients [21]. Therefore, PA is recommended as a healthy lifestyle strategy for the overall brain health of older populations [22]. Likewise, lower-body muscle strength is inversely correlated with depressive symptoms in older adults with depression [23] and nursing home residents with dementia [24]. Lower-body muscle strength is also positively correlated with cognitive function in older Finnish adults [25], older American adults [26], and older Korean adults [27].

PA is defined as any bodily movement of skeletal muscles that results in substantial energy expenditure. Muscle strength is a component of muscular fitness [28], and it is primarily determined by genetics and secondarily by environmental factors such as PA and nutrition [29]. PA and muscle strength are two independent behavioral factors influencing mental health. Some individuals who meet PA recommendations may also have good muscular strength, and vice versa, implying the clinical importance of considering both. Yet, little is known about the combined effect of PA and muscle strength on mental health in older adults. In the current study, involving a representative sample of older Korean adults, we hypothesized that PA combined with muscle strength acts as a moderator in determining the relationship between late-life depression and cognitive function—being more effective than PA or muscle strength alone.

## 2. Materials and Methods

### 2.1. Data Source and Study Participants

The data used in the present study were obtained from the 2020 Korea Longitudinal Study on Aging (KLoSA), a nationwide population-based survey conducted in Korea in which data were collected using a computer-assisted personal interviewing protocol. In brief, a total of 10,097 older adults (6062 females and 4035 males) ≥ 65 years of age participated in the survey. Individuals without a score on the Korean version of the Geriatric Depression Scale Short-Form (K-GDS-SF) and/or unavailable cognition data (*n* = 212) were excluded. The remaining 9885 individuals (3955 males and 5930 females) were used for the final data analyses (Figure 1). The Institutional Review Board of the Korea Institute for Health and Social Affairs reviewed and approved the survey (approval no. 2020-36). Informed consent was obtained from all participants. Detailed information regarding the KLoSA is available through the national public database (https://survey.keis.or.kr/eng/myinfo/login.jsp (accessed on 4 March 2022).

### 2.2. Measured Variables

#### 2.2.1. Cognitive Function 

Cognitive function was assessed using the Korean version of the Mini-Mental Status Examination (K-MMSE) optimized for screening dementia (MMSE-DS) [30]. The MMSE-DS is an updated and validated version of both the Korean version of the MMSE (MMSE-KC) in the Consortium to Establish a Registry for Alzheimer’s disease Assessment Packet (MMSE-KC) and the Seoul Neuropsychological Screening Battery (SNSB) [30]. The maximum total K-MMSE score is 30 points. Age, sex, and education-specific cutoff scores are used to assess mild cognitive impairment (MCI) [30]. 

#### 2.2.2. Assessment of Depression 

Depression was diagnosed using the K-GDS-SF. Geriatric depression was defined as a score ≥ 8 on the K-GDS-SF, physician-diagnosed depression, or the taking of anti-depressant medication(s). A cutoff score of 8 for screening mental illness was assessed and validated in older Korean adults [31,32].

#### 2.2.3. Physical Activity and Lower-Body Muscle Strength 

PA was assessed using a self-reported questionnaire with the question “do you participate in any PA lasting for at least 10 min per session?” If the respondent say “yes”, they were further asked to report the frequency and duration of weekly PA. The total volume of weekly PA was then calculated based on duration (minutes per session) and frequency (days per week). PA was then categorized as sufficient (≥150 min per week) or insufficient (<150 min per week or no PA) based on the global recommendations on PA [33].

Although handgrip strength is one of the most frequently used measurements of muscle strength, especially in geriatric populations, lower limb muscle strength is a better indicator of overall muscle strength [34] and geriatric syndromes [35]. In the current study, a modified sit-to-stand test (STST) was used to evaluate lower-body muscle strength [36]. In brief, the participants were instructed to stand from a sitting position on a chair with both arms folded across the chest 5 times as fast as possible. The performance was scored by completeness (1 = completed successfully, 2 = tried but failed to complete, 3 = could not perform at all). For this study, completed successfully was categorized as a good STST, and tried but failed to complete and could not perform at all were combined and categorized as a poor STST. The validity and reliability of the sit-to-stand test for the assessment of lower-body strength were previously tested and reported in a representative sample of Korean elderly persons [37]. Finally, in order to assess the combined effects of PA and lower-body muscle strength on mental health, PA and STST were then combined and classified as sufficient PA and good STST, or as either insufficient physical activity or poor STST, or as insufficient physical activity and poor STST.

#### 2.2.4. Covariates

The covariates included in the study were age (years), gender (male vs. female), body mass index (BMI), educational level (≤elementary, middle/high school, ≥college), smoking status (current/past smoker, non-smoker), alcohol intake (0, 1–6, ≥7 times/week), and comorbidity. Comorbidity was determined using the diagnoses reported by a doctor of at least one of 16 selected, previously reported chronic conditions [8]. 

### 2.3. Statistical Analyses 

Data distribution normality and multi-collinearity were verified using quantile-quantile plots and variance of inflation factor, respectively. Student’s *t*-test and chi-square test were used to compare continuous and categorical variables, respectively, between MCI-based subgroups. Multivariate logistic regression was used to estimate odds ratios (ORs) and 95% confidence intervals (CIs) of PA and lower-body muscle strength for MCI. Moderation analyses of PA and/or lower-body muscle strength (moderator, W) on the relationship between depression (categorical, X) and cognitive function (continuous, Y) were conducted based on the moderation paths proposed by Baron and Kenny [38], as shown in Figure 2. The Andrew Hayes’ PROCESS macro Modeling 1 in SPSS-PC version 27.0 (IBM Corporation, Armonk, NY, USA) was used to carry out the moderation analysis. For this simple moderation model, the process macro automatically centers the variables, computes the interaction term, runs the regression model with the interaction term, and then tests the simple slopes. The statistical significance of the model was assessed with bias-corrected bootstrapping (*n* = 10,000) and 95% CIs. A detailed explanation of the PROCESS macro for a moderation analysis is provided elsewhere [39]. All other statistical significances were evaluated at *α* = 0.05 using the SPSS-PC version 27.0 (IBM Corporation).

## 3. Results

Table 1 shows the descriptive statistics of the study participants. The overall prevalence of MCI in the study population was 33.5%, with a higher rate in males than in females (35% vs. 32.5%, *p* = 0.013). Individuals with MCI were older (*p* < 0.001), had lower educational levels (*p* < 0.001), smoked less (*p* = 0.026), consumed less alcohol (*p* = 0.005), had more multi-morbidities (*p* < 0.001), were more depressed (*p* < 0.001), less physically active (*p* < 0.001), and had less lower-body muscle strength (*p* < 0.001) compared with individuals without MCI. 

Table 2 shows the bivariate correlations between cognitive function and the measured parameters. Cognitive function was inversely associated with age (*p* < 0.001), gender (*p* = 0.034), marital status (*p* = 0.025), smoking status (*p* = 0.004), and depression (*p* < 0.001), and positively correlated with educational level (*p* < 0.001), PA (*p* < 0.001), and lower-body muscle strength (*p* < 0.001).

Table 3 shows the ORs and 95% Cis of MCI based on depression status and the level of physical activity and lower-body muscle strength. Depressed individuals were at increased risk of MCI (OR = 1.937 and 95% CI = 1.685–2.228, *p* < 0.001) compared with non-depressed individuals (OR = 1.000). The increased OR for MCI remained statistically significant (OR = 1.845 and 95% CI = 1.580–2.154, *p* < 0.001) even after adjusting for all covariates. Individuals who had insufficient PA and poor STST and individuals who had insufficient PA or poor STST were at increased risk of MCI (OR = 1.329 and 95% CI = 1.209–1.46, *p* < 0.001 and OR = 2.822 and 95% CI = 2.488–3.200, *p* < 0001, respectively) compared with individuals who had sufficient PA and good STST (OR = 1.000).

Table 4 shows a moderation effect of PA combined with lower-body muscle strength (W) on the relationship between depression (X) and cognitive function (Y). A significant moderation effect of PA combined with lower-body muscle strength on the relationship between late-life depression and cognitive function was observed (β = −0.2658; 95% CI = −0.3242 to −0.2075). The moderating effect remained statistically significant (β = −0.2644; 95% CI = −0.3256 to −0.2033) even after adjusting for all the covariates. The interaction was further investigated to better understand the moderating effect of physical activity combined with lower-body muscle strength on the relationship between late-life depression and cognitive function. As shown in Figure 3, the strength of the inverse relationship between late-life depression and cognitive function was incremental in the order of sufficient PA and good STST, insufficient PA or poor STST, and insufficient and poor STST. In addition, there was a significant moderating effect of lower-body muscle strength itself (interaction coefficient and SE = −0.5507 and 0.5033, *p* = 0.005)—but not PA—on the relationship between depression and cognitive function.

## 4. Discussion

In this population-based cross-sectional study, the relationship of PA combined with lower-body muscle strength and depression with cognitive function was examined in older Korean adults. Late-life depression, insufficient PA, and poor lower-body muscle strength were significantly associated with an increased risk of MCI. To the best of our knowledge, this is the first study to report that PA combined with lower-body muscle strength is a better moderator in determining the relationship between late-life depression and cognitive function as compared to PA or lower-body muscle strength alone.

The findings of the present study are in an agreement with previous studies in which a moderating effect of PA or muscle strength on the association between late-life depression and cognitive decline has been reported. Hu et al. [40] analyzed the data obtained from 2604 adults ≥ 60 years of age participating in the National Health and Nutrition Examination Survey (2011–2014). In that study, they showed that depressive symptoms were significantly correlated with poor cognitive performance among those who had no or insufficient PA. However, the inverse relationship between depressive symptoms and cognitive function was not statistically significant among those who had sufficient PA (i.e., ≥150 min per week of moderate-to-vigorous PA). Yuan et al. [41] conducted a serial mediation analysis to examine the associations between sleep quality, depression, PA, and cognitive function in 3230 older Chinese adults ≥ 60 years of age. The authors showed that PA frequency was associated with better cognitive function in conjunction with improved sleep quality and lower depressive symptoms.

The cognitive benefit of lower-body muscle strength observed in the current study can also be attributed to its anti-depressant effect. By analyzing the data obtained from 1508 older participants aged 60–85 years in the 1999–2002 National Health and Nutrition Examination Survey, Frith and Loprinzi [26] showed that lower-extremity muscle strength was independently associated with higher cognitive performance. Similarly, we previously analyzed the data obtained from 9920 older adults ≥ 65 years of age in the 2020 KLoSA and showed that muscle strength modulated the relationship between nutritional health risk and depression [42]. 

Late-life depression is closely associated with cognitive impairment in geriatric populations [32,33]. In a follow-up study involving 90 older Chinese adults, Wu et al. [43] showed some evidence that late-life depression might have a unidirectional relationship with cognitive impairment. The authors assessed depressive symptoms and cognition at baseline and 1-year follow-up and showed that depressive symptoms at baseline predicted cognitive decline, but not vice versa—indicating that depression is a potential risk factor for cognitive impairment. By conducting a 7-year prospective cohort study involving 17,596 Chinese adults aged 45 years and older, Bao et al. [44] showed that along with handgrip strength, five-times sit-to-stand test performance was an independent predictor of depression. Taken together, the findings of the current and previous studies suggest that the cognitive benefit of PA combined with muscle strength is associated with its anti-depressant effect, although the underlying mechanisms remain to be elucidated.

In our previous study involving 10,245 older Korean adults who participated in the 2008 baseline survey of the living profiles of older people survey, depression was shown to mediate the negative effects of PA on cognitive function [45]. Similarly, Yuenyongchaiwat et al. [46] conducted a cross-sectional study involving older adults with and without cognitive impairments and found that low PA and high depressive symptoms were associated with an increased risk of cognitive impairment. Furthermore, the mental health benefits of PA and/or muscle strength have been reviewed and well-summarized in previous reviews or meta-analysis studies [47,48].

Several explanations are possible for the cognitive benefit of PA and muscle strength. First, regular PA correlates with several health benefits, including glucose homeostasis, fatty oxidation, improved cardiovascular function, and an enhanced neuro-immune system, as well as better brain health and a reduced risk of dementia and cognitive impairment [16]. Second, participating in PA improves the cerebral cortex and increases blood supply to the associated areas of the brain, which aids brain processing speeds [16] and reduces inflammatory responses associated with the deterioration of brain function [16]. Third, regular PA increases hippocampal neurogenesis in conjunction with neurochemicals such as brain-derived neurotrophic factors and insulin-like growth factors [17], improving neural plasticity and reducing cognitive impairment levels [18]. Fourth, the maintenance of normal muscle strength counteracts or minimizes the risk of late-life depression by maintaining physical independence, resulting in fewer difficulties performing activities of daily living and lowering the risk of physical disabilities [49]. Fifth, muscle mass and strength exert an antidepressant effect by inducing anti-inflammatory responses while suppressing inflammatory responses [50].

The present study had several limitations. First, the association between PA combined with muscle strength and mental health may be bidirectional [51]. A well-designed intervention study is necessary to delineate the complex nature of the relationship between PA and muscle strength and mental health. Second, the cross-sectional nature of the study limits the interpretation of the current findings in a cause-and-effect manner. Third, self-reported PA is likely to be subject to somewhat individual variations and has a weak or moderate relationship with objectively measured PA [52]. Accelerometer-based objective assessment of PA would be more appropriate to determine the exact role of PA in terms of mental health in older adults. 

## 5. Conclusions

In summary, the association between PA, muscle strength, depression, and cognitive function was investigated in a representative sample of older Korean adults. The current findings showed that PA combined with lower-body muscle strength is a better moderator in determining the relationship between depression and cognitive function, indicating that promotion of regular PA in conjunction with maintenance of muscle strength should be recommended as a healthy aging strategy for reducing the risk of mental illnesses.

## Figures and Tables

**Figure 1 ijerph-19-08769-f001:**
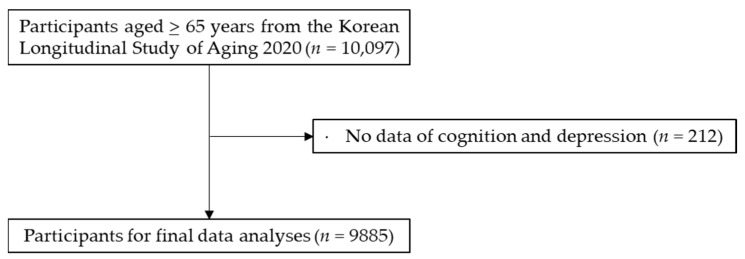
A flow chart for the selection of study participants.

**Figure 2 ijerph-19-08769-f002:**
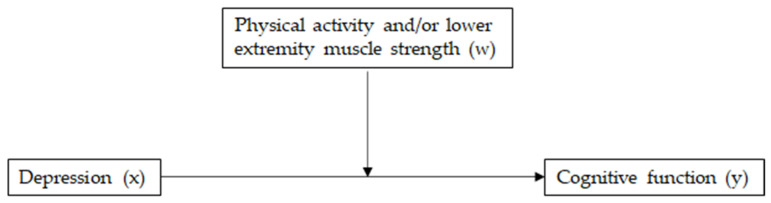
A conceptual diagram of the relationship between depression (x) and cognitive function (y) moderated by physical activity and/or lower-body muscle strength (w).

**Figure 3 ijerph-19-08769-f003:**
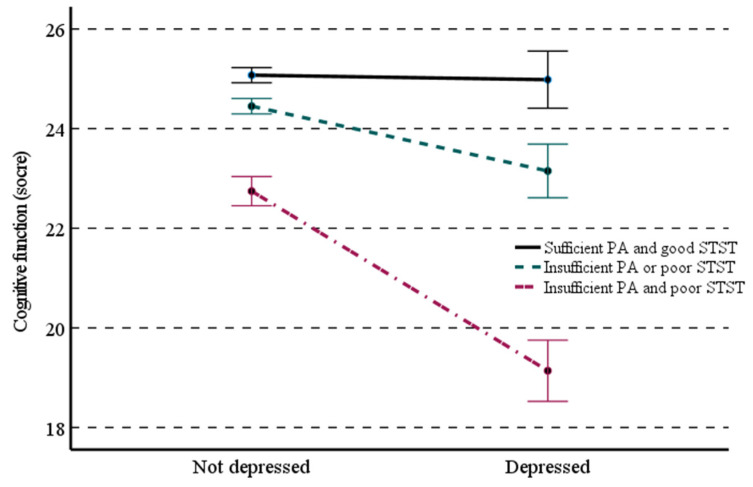
Effect of physical activity (PA) combined with sit-to-stand test (STST) on the relationship between depression and cognitive function in Korean older adults. STST was classified as good if the sit-to-stand test consecutively performed 5 times or poor if the 5-times sit-to-stand test was failed/not attempted. PA was classified as sufficient if meeting the global recommendations on PA (≥150 min per week) or insufficient if not meeting the global recommendations on PA (<150 min per week or no PA).

**Table 1 ijerph-19-08769-t001:** Descriptive statistics of study participants.

Variables	Normal (n = 6572/66.5%)	MCI (n = 3313/33.5%)	Total (n = 9885/100%)	ES	*p* Value
Age group, (years)	73.0 ± 6.4	74.3 ± 6.7	73.4 ± 6.5	0.120	<0.001
Body mass index (kg/m^2^)	23.6 ± 2.7	23.5 ± 2.7	23.6 ± 2.6	0.002	0.092
Gender, n (%)				0.025	0.013
Male	2572 (39.1)	1383 (41.7)	3955 (40.0)		
Female	4000 (60.9)	1930 (58.3)	5930 (60.0)		
Marriage				0.017	0.092
Never married	29 (0.4)	12 (0.4)	41 (0.4)		
Married with spouse	3910 (59.5)	1917 (57.9)	5827 (58.9)		
Married without spouse	2633 (40.1)	1384 (41.8)	4017 (40.6)		
Educational background, n (%)				0.084	<0.001
Elementary or less	3058 (46.5)	1355 (40.9)	4413 (44.6)		
Middle/high school	3125 (47.6)	1843 (55.6)	4968 (50.3)		
College or higher	389 (5.9)	115 (3.5)	504 (5.1)		
Smoking status, n (%)				0.022	0.026
Never smoked	5822 (88.6)	2984 (90.1)	8806 (89.1)		
Current/past smokers	750 (11.4)	329 (9.9)	1079 (10.9)		
Alcohol intake (times/week)				0.033	0.005
0	5593 (85.1)	2899 (87.5)	8492 (85.9)		
1–6	905 (13.8)	381 (11.5)	1286 (13.0)		
≥7	74 (1.1)	33 (1.0)	107 (1.1)		
Multi-morbidity, n (%)				0.076	<0.001
None	1184 (20.9)	493 (17.8)	1677 (19.9)		
Single	2000 (35.3)	915 (33.0)	2915 (34.5)		
Multiple	2483 (43.8)	1366 (49.2)	3849 (45.6)		
Depression, n (%)				0.095	<0.001
Undepressed	6115 (93.0)	2894 (87.4)	9009 (91.1)		
Depressed	457 (7.0)	419 (12.6)	876 (8.9)		
Physical activity, n (%)				0.066	<0.001
Active	3588 (54.6)	1576 (47.6)	5164 (52.2)		
Inactive	2984 (45.4)	1737 (52.4)	4721 (47.8)		
Lower-body muscle strength, n (%)					<0.001
Good STST	5162 (82.5)	2094 (66.1)	7256 (77.0)	0.183	
Poor STST	1097 (17.5)	1073 (33.9)	2170 (23.0)		
PA and lower-body muscle strength				0.170	<0.001
Sufficient PA and good STST	3035 (48.5)	1157 (36.5)	4192 (44.5)		
Insufficient PA or poor STST	2563 (40.9)	1299 (41.0)	3862 (41.0)		
Insufficient PA and poor STST	661 (10.6)	711 (22.5)	1372 (14.6)		

BMI: body mass index; MCI: mild cognitive impairment. STST: sit-to-stand test; PA: physical activity. Effect sizes (ES) were calculated using Cohen’s d and Cramer’s v for categorical and continuous variables, respectively. STST was classified as good if the sit-to-stand test was consecutively performed 5 times or poor if the 5-times sit-to-stand test was failed/not attempted. PA was classified as sufficient if meeting the global recommendations on PA (≥150 min per week) or insufficient if not meeting the global recommendations on PA (<150 min per week or no PA).

**Table 2 ijerph-19-08769-t002:** Linear regression for determinants of cognitive function.

Variables	Beta	95% CI	r^2^_part_	*p* Value	VIF
Age	−0.182	−0.200–−0.165	−0.200	<0.001	1.460
Gender	−0.261	−0.501–−0.020	−0.021	0.034	1.529
Marriage	−0.247	−0.463–−0.031	−0.023	0.025	1.279
Body mass index	0.032	−0.005–0.069	0.017	0.092	1.026
Education	1.282	1.090–1.474	0.131	<0.001	1.387
Smoking	−0.484	−0.816–−0.152	−0.029	0.004	1.182
Alcohol intake	0.156	−0.056–0.368	0.015	0.148	1.270
Multi-comorbidity	−0.098	−0.232–0.036	−0.014	0.153	1.129
Depression	−2.591	−2.955–−2.226	0.019	<0.001	1.000
PA	1.482	1.690–1.275	0.019	<0.001	1.000
LBMS	4.215	4.452–3.977	0.114	<0.001	1.000

CI: confidence interval; VIF: variance inflation factor; PA: physical activity; LBMS: lower-body muscle strength.

**Table 3 ijerph-19-08769-t003:** Odds ratios (ORs) and 95% confidence intervals (CIs) of mild cognitive impairment (MCI) by depression and physical activity (PA) combined with sit-to-stand test (STST).

Predictors	Model 1		Model 2	
	OR (95% CI)	*p* Value	OR (95% CI)	*p* Value
Depressional status				
Not depressed	1 (reference)		1 (reference)	
Depressed	1.937 (1.685–2.228)	<0.001	1.845 (1.580–2.154)	<0.001
PA and STST				
Sufficient PA and good STST	1 (reference)		1 (reference)	
Insufficient PA or poor STST	1.329 (1.209–1.462)	<0.001	1.347 (1.214–1.496)	<0.001
Insufficient PA and poor STST	2.822 (2.488–3.200)	<0.001	2.437 (2.356–3.230)	<0.001

Model 1 unadjusted. Model 2 adjusted for age, gender, education, smoking, alcohol intake, and comorbidities. STST was classified as good if the sit-to-stand test was consecutively performed 5 times or poor if the 5-times sit-to-stand test was failed/not attempted. PA was classified as sufficient if meeting the global recommendations on PA (≥150 min per week) or insufficient if not meeting the global recommendations on PA (<150 min per week or no PA).

**Table 4 ijerph-19-08769-t004:** A moderation analysis of physical activity (PA) combined with lower-body muscle strength (LBMS) for the relationship between depression status and cognitive function.

Predictors	Coefficients	SE	t	*p*	95% CI	
					Lower	Upper
Model 1 (R^2^ = 0.106, F = 371.681, *p* < 0.001)						
Depression status	2.8621	0.6985	4.0977	<0.001	1.4930	4.2312
PA combined with LBMS	−0.3418	0.2674	−1.2783	0.2012	−0.8659	0.1823
Interaction	−1.6350	0.2316	−7.0599	<0.001	−2.0890	−1.1811
R^2^ change due to the moderator = 0.005 (F = 49.842, *p* < 0.001)						
Model 2 (R^2^ = 0.132, F = 58.830, *p* < 0.001)						
Depression status	2.2930	1.1465	1.9999	0.0450	0.0450	4.5409
PA combined with LBMS	0.4615	0.4618	0.9994	0.3177	−0.4439	1.3669
Interaction	−1.3923	0.3867	−3.6002	0.0003	−2.1505	−0.6341
R^2^ change due to the moderator = 0.003 (F = 12.962, *p* = 0.003)						

Model 1 unadjusted. Model 2 adjusted for age, gender, education, smoking, alcohol intake, and comorbidities. SE: standard error; CI: confidence interval.

## Data Availability

Data are accessible via the National public database (https://survey.keis.or.kr/eng/myinfo/login.jsp/ accessed 10 June 2021).

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
