# Peer review of "The Relationship between Late-Life Depression and Cognitive Function in Older Korean Adults: A Moderation Analysis of Physical Activity Combined with Lower-Body Muscle Strength"

_ijerph, 2022, doi:10.3390/ijerph19148769_

Round 1
Reviewer 1 Report
This manuscript, entitled “Physical Activity Combined with Lower Body Muscle Strength Modulates the Effects of Late-2 Life Depression on Cognitive Function in Older Korean Adults”, investigated the relationship between late-life depression and cognition in Korean aging population and examined the moderation of physical activity and lower body muscle strength. This study enrolled a large sample of participants and found that physical activity combined with lower body muscle strength would play a significant moderating role in this relationship of interest. The manuscript was well written, and this study added novel findings into the existing evidence. However, before this manuscript could be accepted for publication, there are some comments which need to be addressed.
Introduction
1. My major concern is that, although cognitive benefits of PA and muscle strength were addressed, the rationale and mechanisms may be missed why the authors believed that PA combined with lower body muscle strength may moderate the relationship between depression and cognition.
2. It is also not clear why this study decided to only measure lower body muscle strength, specifically when the authors cited many studied which measured handgrip strength to support their findings. The rationale needs to be addressed.
3. Line 27: please delete “(who)”
4. Line 37: which will be likely to increase…
5. Line 43: [11]
6. Line 46-47: The reference(s) should be provided.
Methods
7. I would suggest the authors providing information regarding reliability and validity for each measurement.
8. I am concerned about the definition for physical activity. Based on the description, the participant was identified as “be physically active” if they answered YES to the question, indicating they engaged in >10 min of PA per session. However, this amount of PA was much lower than that which was recommended by the WHO or other organization (i.e., at least 150 min of MVPA per week). Is there any evidence for this definition?
9. As there was no significant difference in BMI between two groups, why was it defined as covariate?
10. According to Hayes, corrected bootstrap procedures with 10,000 simulations were recommended. Please refer to: Hayes, A.F.; Scharkow, M. The relative trustworthiness of inferential tests of the indirect effect in statistical mediation analysis: Does method really matter? Psychol. Sci. 2013, 24, 1918–1927
11. Line 74: validated version
Results
12. Table 2 and 3: PA combined with MS; PA combined with LBMS -> please be consistent
13. Line 150: depression (X) and cognitive function (Y) -> please be consistent with the section of statistical analysis 
Discussion
14. As mentioned above, the authors cited studies measuring handgrip strength to support their findings (i.e., Ref 21, 29, 34, and 45). If the authors were looking for supporting evidence, other studies measuring lower body strength would be needed.
15. Line 205-214: When this paragraph was discussing the mediation of depression on the relationship between PA and cognition, this may be irrelevant to the findings of this study. I would suggest to remove this paragraph.
16. Line 237: summary
Author Response
In our Response the Comments/Critics by Reviewer#1
Thanks for the thoughtful comments and critics for the quality of the manuscript. We did our best to address the comments/critics point-by-point and highlighted in yellow color here and in the text.
Introduction
Q1) My major concern is that, although cognitive benefits of PA and muscle strength were addressed, the rationale and mechanisms may be missed why the authors believed that PA combined with lower body muscle strength may moderate the relationship between depression and cognition.
ANS1) Thanks for the comments. Although we accept the critics, this is a cross-sectional study. We do not want to exaggerate the biological mechanism(s) by which physical activity combined with muscle strength modulates the relationship between depression and cognition. Rather than, the following rationales are added in our response to the comments.
“Physical activity (PA) is positively correlated with brain processing speed [16], hippocampal neurogenesis [17], and neural plasticity [18], providing a protective effect against mental illnesses such as late-life depression, age-related cognitive decline, and dementia [17]. The cognitive benefits of PA were reported in studies involving older Chinese [19] and Swedish adults [20]. The anti-depressant effects of PA are also summarized in clinically depressed patients [21]. Therefore, PA is recommended as a healthy lifestyle strategy for the overall brain health of older populations [22]. Likewise, lower body muscle strength is inversely correlated with depressive symptoms in older adults with depression [23] and nursing home residents with dementia [24]. Lower body muscle strength is also positively correlated with cognitive function in older Finnish adults [25], American older adults [26], and older Korean adults [27].”
“PA is defined as any bodily movement of skeletal muscles that results in substantial energy expenditure. Muscle strength is a component of muscular fitness [28], and it is primarily determined by genetics and secondarily by environmental factors such as PA and nutrition [29]. PA and muscle strength are two independent behavioral factors influencing mental health. Some individuals who meet the PA recommendations may also have good muscular strength, and vice versa, implying the clinical importance of considering both. Yet, little is known about the combined effect of PA and muscle strength on mental health in older adults. In the current study involving a representative sample of older Korean adults, we hypothesized that PA combined with muscle strength modulates the relationship between late-life depression and cognitive function better than PA or muscle strength alone.”
Q2) It is also not clear why this study decided to only measure lower body muscle strength, specifically when the authors cited many studied which measured handgrip strength to support their findings. The rationale needs to be addressed.
ANS2) Thanks for the comment. In our response to the comment, the following explanation is given to Methods (2.2.3. Physical activity and lower body muscle strength):
“Although handgrip strength is one of the most frequently used measurements of muscle strength, especially in geriatric populations, lower limb muscle strength is a better indicator of overall muscle strength [34] and geriatric syndrome [35]. In the current study, a modified sit-to-stand test (STST) was used to evaluate lower body muscle strength [36].”
Q3) Line 27: please delete “(who)”
ANS3) Thanks. We removed it as requested.
Q4) Line 37: which will be likely to increase…
ANS4) Thanks. We corrected the sentence as suggested.
Q5) Line 43: [11]
ANS5) Thanks. Corrected as suggested.
Q6) Line 46-47: The reference(s) should be provided.
ANS6) Thanks. Lines 46-47 are removed because of modifications/corrections.
Methods
Q7) I would suggest the authors providing information regarding reliability and validity for each measurement.
ANS7) Thanks. The reliability and validity of each measurement are cited using relevant references.
- The MMSE-DS is an updated and valid version of both the Korean version of MMSE (MMSE-KC) in the Consortium to Establish a Registry for Alzheimer’s disease Assessment Packet (MMSE-KC) and the Seoul Neuropsychological Screening Battery (SNSB) [23].
- The Korean version of the Geriatric Depression Scale Short-Form (K-GDS-SF) was assessed and validated in previous studies involving Korean geriatric populations [24, 25].
- Physical activity was categorized as active (≥150 min per week) or inactive (<150 min per week) based on the global recommendations on physical activity (World Health Organization, 2010).
- The validity and reliability of the sit-to-stand test for the assessment of lower body strength were previously tested and reported in a representative sample of Korean elderly persons (Nam and Kim, 2019).
Q8) I am concerned about the definition for physical activity. Based on the description, the participant was identified as “be physically active” if they answered YES to the question, indicating they engaged in >10 min of PA per session. However, this amount of PA was much lower than that which was recommended by the WHO or other organization (i.e., at least 150 min of MVPA per week). Is there any evidence for this definition?
ANS8) Sorry for this confusion. Classification of physical activity was made according to the global recommendations on physical activity. Description of physical activity assessment and classification is revised as follows;
“PA was assessed using a self-reported questionnaire with the question “do you participate in any PA lasting for at least 10 minutes per session?” If the respondent say “yes”, they were further asked to report the frequency and duration of weekly PA. The total volume of weekly PA was then calculated based on duration (minutes per session) and frequency (days per week). PA was then categorized as sufficient (≥150 min per week) or insufficient (<150 min per week or no PA) based on the global recommendations on PA [33].”
Q9) As there was no significant difference in BMI between two groups, why was it defined as covariate?
ANS9) Thanks for the comments. BMI and marriage were excluded as covariates in Tables 3 and 4, although the outcomes are the same.
Q10) According to Hayes, corrected bootstrap procedures with 10,000 simulations were recommended. Please refer to: Hayes, A.F.; Scharkow, M. The relative trustworthiness of inferential tests of the indirect effect in statistical mediation analysis: Does method really matter? Psychol. Sci. 2013, 24, 1918–1927.
ANS10) Thanks. In our response to the comment. We re-run the moderation analysis with corrected bootstrap with 10,000 simulations. The results in Table 4 are revised accordingly.
Q11) Line 74: validated version
ANS11) Thanks. Corrected as suggested.
Results
Q12). Table 2 and 3: PA combined with MS; PA combined with LBMS -> please be consistent
ANS12) Thanks. MS is corrected as LBMS.
Q13) Line 150: depression (X) and cognitive function (Y) -> please be consistent with the section of statistical analysis 
ANS13) Thanks. Depression and cognitive function are consistently denoted as X and Y, respectively.
Discussion
Q14) As mentioned above, the authors cited studies measuring handgrip strength to support their findings (i.e., Ref 21, 29, 34, and 45). If the authors were looking for supporting evidence, other studies measuring lower body strength would be needed.
ANS14) Thanks. References of 21, 29, and are replaced with other references.
- Frith E, Loprinzi PD. The Association between Lower Extremity Muscular Strength and Cognitive Function in a National Sample of Older Adults. J Lifestyle Med. 2018 Jul;8(2):99-104.
- Zhai F, Liu J, Su N, Han F, Zhou L, Ni J, Yao M, Zhang S, Jin Z, Cui L, Tian F, Zhu Y. Disrupted white matter integrity and network connectivity are related to poor motor performance. Sci Rep. 2020 Oct 27;10(1):18369.
- Bao, M.; Chao, J.; Sheng, M.; Cai, R.; Zhang, N.; Chen, H. Longitudinal association between muscle strength and depression in middle-aged and older adults: a 7-year prospective cohort study in China. Affect. Disord. 2022, 301, 81-86.
Q15) Line 205-214: When this paragraph was discussing the mediation of depression on the relationship between PA and cognition, this may be irrelevant to the findings of this study. I would suggest to remove this paragraph.
ANS15) Thanks. Lines 205-214 and reference 45 are removed as suggested.
Q16) Line 237: summary
ANS16) Thanks. It is corrected as “summary”.

Reviewer 2 Report
The authors are examining the association among cognitive function, depression, physical activity and lower body muscle strength among older adults. Strengths of the manuscript include a large sample size (n=10,097), a functional measure of strength, and representation of both males and females. The manuscript will be stronger with a few clarifications.
Major comments
The paper is based on the idea that previous studies have not looked at the combination of muscle strength and physical activity. The introduction does not develop the need to examine muscle strength as a moderator in conjunction with physical activity. Perhaps some of the discussion should be moved to the introduction to better introduce the idea. The analyses do not allow the reader to see any benefit of adding muscle strength. Consider analyses that can demonstrate the benefit of adding muscle strength to the analyses. Is the combination of physical activity and muscle strength a better moderator than physical activity alone? This is a significant limitation of the paper. This needs greater clarification in order the review the results and discussion.
It is not clear how physical activity and muscle strength are “added”. Because these factors were combined, it is difficult to interpret the results. In figure 3 the authors report categories of active and high strength, inactive or lower strength, both inactive and low strength. Better justification is needed for these categories.
There is significant literature frailty and how to define the concept. The authors should be more specific indicating poor scores on the strength test indicate poor lower body strength.
Greater details are needed regarding the moderator analyses. The authors indicate “Andrew Hayes’ PROCESS macro” but do not provide a reference. Greater detail is needed regarding how this particular technique demonstrates moderation. Specifically, how does a single regression with an interaction term indicate the moderation of one variable on the other? This is a significant limitation of the paper. This needs greater clarification in order the review the results and discussion.
Table 4 does not provide information stratified on levels of physical activity and muscle strength (lines 150-151).
The discussion reviews literature but does not compare the results from this paper to the results from other paper.
The authors use the terms elderly and geriatric. The manuscript would be strengthened by using a single term consistently.
Line 184. You cannot demonstrate an effect of physical activity on mental health using NHANES data. Please check the reference and ensure it is appropriately described.
Minor comments
Lines 26-27, are the authors suggesting that decreased fertility results in geriatric conditions. Consider revising the sentence.
Lines 36-38 Will the prevalence of dementia among older adults increase or will the impact of dementia increase as the population ages?
Line 41 change devastating to significant.
Author Response
In our Response the Comments/Critics by Reviewer#2
Thanks for the thoughtful comments and critics for the quality of the manuscript. We did our best to address the comments/critics point-by-point and highlighted in yellow color here and in the text.
Major comments
Q1) The introduction does not develop the need to examine muscle strength as a moderator in conjunction with physical activity. Perhaps some of the discussion should be moved to the introduction to better introduce the idea.
ANS1) Thanks for the comments. the rationale of the study in Introduction is revised as follows:
“Physical activity (PA) is positively correlated with brain processing speed [16], hippocampal neurogenesis [17], and neural plasticity [18], providing a protective effect against mental illnesses such as late-life depression, age-related cognitive decline, and dementia [17]. The cognitive benefits of PA were reported in studies involving older Chinese [19] and Swedish adults [20]. The anti-depressant effects of PA are also summarized in clinically depressed patients [21]. Therefore, PA is recommended as a healthy lifestyle strategy for the overall brain health of older populations [22]. Likewise, lower body muscle strength is inversely correlated with depressive symptoms in older adults with depression [23] and nursing home residents with dementia [24]. Lower body muscle strength is also positively correlated with cognitive function in older Finnish adults [25], American older adults [26], and older Korean adults [27].”
“PA is defined as any bodily movement of skeletal muscles that results in substantial energy expenditure. Muscle strength is a component of muscular fitness [28], and it is primarily determined by genetics and secondarily by environmental factors such as PA and nutrition [29]. PA and muscle strength are two independent behavioral factors influencing mental health. Some individuals who meet the PA recommendations may also have good muscular strength, and vice versa, implying the clinical importance of considering both. Yet, little is known about the combined effect of PA and muscle strength on mental health in older adults. In the current study involving a representative sample of older Korean adults, we hypothesized that PA combined with muscle strength modulates the relationship between late-life depression and cognitive function better than PA or muscle strength alone.”
Q2) The analyses do not allow the reader to see any benefit of adding muscle strength. Consider analyses that can demonstrate the benefit of adding muscle strength to the analyses. Is the combination of physical activity and muscle strength a better moderator than physical activity alone? This is a significant limitation of the paper. This needs greater clarification in order the review the results and discussion.
ANS2) Thanks for the comments. We re-rune the moderation analyses using PA, lower body muscle strength, and PA combined with lower body muscle strength as a separate moderator and found the following results; PA alone was not a significant moderator in determining the relationship between depression and cognitive function (interaction coefficient = -0.4315, p=0.4567), while MM was a significant moderator in determining the relationship between depression and cognitive function (interaction coefficient = -0.5507, p=0.005). However, PA and MM were additive as a moderator (coefficient = -1.3923, p=0.003). In our response to the comment, therefore, we added the following statement to Results.
“In addition, there was a significant moderation effect of lower body muscle strength itself (interaction coefficient and SE = -0.5507 and 0.5033, p=0.005), but not PA, on the relationship between depression and cognitive function (data not shown).”
Q3) It is not clear how physical activity and muscle strength are “added”. Because these factors were combined, it is difficult to interpret the results. In figure 3 the authors report categories of active and high strength, inactive or lower strength, both inactive and low strength. Better justification is needed for these categories.
ANS3) Thanks for the comments. The WHO recommends at least 150 minutes of moderate-to-vigorous physical activity for adults. The validity and reliability of the 5 times sit-to-stand test for the assessment of overall muscle strength have been well established. Study participants were classified as sufficient PA or insufficient PA based on the WHO physical activity recommendation, and they were also classified as either good STST or poor STST based on the 5 times sit-to-stand test. The PA- and STST-based categories were then combined and classified as sufficient physical activity and good STST (sufficient PA and good STST) or either insufficient physical activity or poor STST (insufficient PA or poor STST) or insufficient physical activity and poor STST (insufficient PA and poor STST).
In our response to the comments, Methods (2.2.3. Physical activity and lower body muscle strength) is entirely revised as follows:
“PA was assessed using a self-reported questionnaire with the question “do you participate in any PA lasting for at least 10 minutes per session?” If the respondent say “yes”, they were further asked to report the frequency and duration of weekly PA. The total volume of weekly PA was then calculated based on duration (minutes per session) and frequency (days per week). PA was then categorized as sufficient (≥150 min per week) or insufficient (<150 min per week or no PA) based on the global recommendations on PA [33].”
“Although handgrip strength is one of the most frequently used measurements of muscle strength, especially in geriatric populations, lower limb muscle strength is a better indicator of overall muscle strength [34] and geriatric syndrome [35]. In the current study, a modified sit-to-stand test (STST) was used to evaluate lower body muscle strength [36]. In brief, the participants were instructed to stand from a sitting position on a chair with both arms folded across the chest 5 times as fast as possible. The performance was scored by completeness (1 = completed successfully, 2 = tried but failed to complete, 3 = could not perform at all). For this study, completed successfully was categorized as good STST, and tried but failed to complete and could not perform at all were combined and categorized as poor STST. The validity and reliability of the sit-to-stand test for the assessment of lower body strength were previously tested and reported in a representative sample of Korean elderly persons [37]. Finally, in order to assess the combined effects of PA and lower body muscle strength on mental health, PA and STST were then combined and classified as sufficient PA and good or either insufficient physical activity or poor STST or insufficient physical activity and poor STST.”
Q4) There is significant literature frailty and how to define the concept. The authors should be more specific indicating poor scores on the strength test indicate poor lower body strength.
ANS4) Thanks for the comments. As mentioned above, we reclassified PA (sufficient vs. insufficient) and lower body muscle strength (good STSTvs. poor STST).
Q5) Greater details are needed regarding the moderator analyses. The authors indicate “Andrew Hayes’ PROCESS macro” but do not provide a reference. Greater detail is needed regarding how this particular technique demonstrates moderation. Specifically, how does a single regression with an interaction term indicate the moderation of one variable on the other? This is a significant limitation of the paper. This needs greater clarification in order the review the results and discussion.
ANS5) Thanks. In our response to the comments, the explanations for the moderation analyses used in the current study are revised as follows;
“Moderation analyses of PA and/or lower body muscle strength (moderator, W) on the relationship between depression (categorical, X) and cognitive function (continuous, Y) were conducted based on the moderation paths proposed by Baron and Kenny (1986). The Andrew Hayes’ PROCESS macro Modeling 1 in SPSS-PC version 27.0 (IBM Corporation, Armonk, NY, USA) was used to carry out the moderation analysis.”
“For this simple moderation model, the process macro automatically centers the variables, computes the interaction term, runs the regression model with the interaction term, and then tests the simple slopes. The statistical significance of the model was assessed with bias-corrected bootstrapping (n = 10,000) and 95% CIs. A detailed explanation of the PROCESS macro for a moderation analysis is provided elsewhere (Hays, 2012). All other statistical significances were evaluated at a = 0.05 using the SPSS-PC version 27.0 (IBM Corporation).”
Q6) Table 4 does not provide information stratified on levels of physical activity and muscle strength (lines 150-151).
ANS6) Thanks. Detailed descriptions about the classifications of physical activity and muscle strength are now added to Table 4 (refer to Table 4 footnotes).
Q7) The authors use the terms elderly and geriatric. The manuscript would be strengthened by using a single term consistently.
ANS7) Thanks. It is consistently corrected as suggested.
Q8) Line 184. You cannot demonstrate an effect of physical activity on mental health using NHANES data. Please check the reference and ensure it is appropriately described.
ANS8) Thanks. The sentences are corrected as follows;
“Hu et al. [40] analyzed the data obtained from 2,604 adults ≥ 60 years of age participating in the National Health and Nutrition Examination Survey (2011–2014). In that study, they showed that depressive symptoms were significantly correlated with poor cognitive performance among those who had no or insufficient PA. However, the inverse relationship between depressive symptoms and cognitive function was not statistically significant among those who had sufficient PA (i.e., >150 minutes per week of moderate-to-vigorous PA).”
Minor comments
Q9) Lines 26-27, are the authors suggesting that decreased fertility results in geriatric conditions. Consider revising the sentence.
ANS9) Thanks. The paragraph is now corrected as follows;
“Aging of the global population is an inevitable trend attributable to a decrease in fertility rate and an increase in life expectancy.”
Q10) Lines 36-38 Will the prevalence of dementia among older adults increase or will the impact of dementia increase as the population ages?
ANS10) Thanks. The paragraph is corrected as follows;
“The prevalence of dementia among elderly Koreans is estimated to be 9.2%, which is higher than Western and other Asian populations [7] and likely to increase due to a rapid rise in older populations [8].”
Line 41 change devastating to significant.
ANS) Corrected as suggested.

Reviewer 3 Report
Thank you for the Authors contribution to this study. The manuscript is interesting, but it needs to be revised. The first part is short and general. It should contain more information about the perspective of the study. Unfortunately, the “material and methods” section is not detailed enough, without it, it's hard to understand the result. Furthermore, the used measures have unknown reliability.
Detailed comments to the authors:
· The introduction is short and superficial. I suggest adding more studies on the effects of physical activity on lower body muscles and depression.
· It would be beneficial to define "late-life".
· Add goals and hypotheses to the end of the introduction.
· Please provide more information on all the measures: the item number, reliability values, etc.
· The big issue of the study is the measures of physical activity. A single question is not a reliable tool for assessing such a complex phenomenon. The WHO distinguishes moderate and vigorous physical activity, which are measured with a complex questionnaire or accelerometer. I suggest using the 'International Physical Activity Questionnaire” to assess physical activity.
Author Response
In our Response the Comments/Critics by Reviewer#3
Thanks for the thoughtful comments and critics for the quality of the manuscript. We did our best to address the comments/critics point-by-point and highlighted in yellow color here and in the text.
Q1) The introduction is short and superficial. I suggest adding more studies on the effects of physical activity on lower body muscles and depression.
Q2) It would be beneficial to define "late-life".
ANS2) Thanks. In our response to the comment, late-life depression is defined as follows;
“Late-life depression is generally defined as depression that occurs for the first time after age 60 (Van Damme A, Declercq T, Lemey L, Tandt H, Petrovic M. Late-life depression: issues for the general practitioner. Int J Gen Med. 2018;11:113-120).”
Q3) Please provide more information on all the measures: the item number, reliability values, etc.
Q3) Thanks. The validity of all the measures (depressive symptoms, cognitive function, the 5 consecutive time sit-to-stand test) is now cited as follows;
- The MMSE-DS is an updated and valid version of both the Korean version of MMSE (MMSE-KC) in the Consortium to Establish a Registry for Alzheimer’s disease Assessment Packet (MMSE-KC) and the Seoul Neuropsychological Screening Battery (SNSB) [23].
- The Korean version of the Geriatric Depression Scale Short-Form (K-GDS-SF) was assessed and validated in previous studies involving Korean geriatric populations [24, 25].
- Physical activity was categorized as active (≥150 min per week) or inactive (<150 min per week) based on the global recommendations on physical activity (World Health Organization, 2010).
- The validity and reliability of the sit-to-stand test for the assessment of lower body strength were previously tested and reported in a representative sample of Korean elderly persons (Nam and Kim, 2019).
Q4) The big issue of the study is the measures of physical activity. A single question is not a reliable tool for assessing such a complex phenomenon. The WHO distinguishes moderate and vigorous physical activity, which are measured with a complex questionnaire or accelerometer. I suggest using the 'International Physical Activity Questionnaire” to assess physical activity.
Q4) Thanks for the comment. A self-reported questionnaire to assess daily PA was used in the Korea Longitudinal Study on Aging (KLoSA). We agree with the weakness of the self-reported PA used in the current study, which is a major study limitation. Yet, we provided a detailed description of PA as follows;
“PA was assessed using a self-reported questionnaire with the question “do you participate in any PA lasting for at least 10 minutes per session?” If the respondent say “yes”, they were further asked to report the frequency and duration of weekly PA. Total volume of weekly PA was calculated based on duration (minutes per session) and frequency (days per week). PA was then categorized as sufficient (≥150 min per week) or insufficient (<150 min per week or no PA) based on the global recommendations on PA [32].”
Additionally, self-reported PA assessment was listed as a study limitation as follows;
“Third, self-reported PA is likely to be subjected to somewhat individual variations and has a weak or moderate relationship with objectively measured PA [51]. Accelerometer-based objective assessment of PA would be more appropriate to determine the exact role of PA in terms of mental health in older adults.”

Round 2
Reviewer 1 Report
The manuscript has been significantly improved, and more vigorous rationales have been provided. I only have few minor comments.
As this study investigated the moderation of PA and STST, I strongly recommended to consistently use the term "moderation" and "moderator" to avoid any confusion, instead of modulate/modulator.
Author Response
Thanks again for the thoughtful comments and critics for the quality of the manuscript. We did our best to address the comments/critics point-by-point and highlighted in yellow color here and in the text.
Q1) As this study investigated the moderation of PA and STST, I strongly recommended to consistently use the term "moderation" and "moderator" to avoid any confusion, instead of modulate/modulator.
QNS1) Thanks for the comment. In our response to the comment, the term “moderation” and “moderator” is consistently used in the revised manuscript.

Reviewer 3 Report
The Authors provided a significant change to their manuscript. It is now suitable for publication.
Author Response
Thanks for the positive comment.